

# Evolutionary analysis of endogenous intronic retroviruses in primates reveals an enrichment in transcription binding sites associated with key regulatory processes

Melissa Calero-Layana[1], Carmen López-Cruz[1], Agustín Ocaña[1], Eduardo Tejera[1,2] and Vinicio Armijos-Jaramillo[1,2]

[1] Ingeniería en Biotecnología. Facultad de Ingeniería y Ciencias Aplicadas, Universidad de las Americas, Quito, Ecuador
[2] Grupo de Bio-Quimioinformática, Universidad de las Americas, Quito, Ecuador

## ABSTRACT

**Background**. Endogenous retroviruses (ERVs) are the result of the integration of retroviruses into host DNA following germline infection. Endogenous retroviruses are made up of three main genes: *gag*, *pol*, and *env*, each of which encodes viral proteins that can be conserved or not. ERVs have been observed in a wide range of vertebrate genomes and their functions are associated with viral silencing and gene regulation.
**Results**. In this work, we studied the evolutionary history of endogenous retroviruses associated with five human genes (*INPP5B, DET1, PSMA1, USH2A,* and *MACROD2*), which are located within intron sections. To verify the retroviral origin of the candidates, several approaches were used to detect and locate ERV elements. Both orthologous and paralogous genes were identified by Ensembl and then analyzed for ERV presence using RetroTector. A phylogenetic tree was reconstructed to identify the minimum time point of ERV acquisition. From that search, we detected ERVs throughout the primate lineage and in some other groups. Also, we identified the minimum origin of the ERVs from the parvorder Catarrhini to the Homininae subfamily.
**Conclusions**. With the data collected, and by observing the transcription factors annotated inside ERVs, we propose that these elements play a relevant role in gene expression regulation and they probably possess important features for tumorigenesis control.

# BACKGROUND

Given that endogenous retroviruses (ERVs) are brought about by the retroviruses' integration into the host's DNA after a germline infection, they are directly transmissible from parents to children, *i.e.,* from one generation to the next (*Johnson, 2015*; *Xu et al., 2018*). ERVs are formed through multiple integrations of exogenous retroviruses throughout the species' evolution (*Lavialle et al., 2013*). Within the vertebrate genomes,

Corresponding author
Vinicio Armijos-Jaramillo,
vinicio.armijos@udla.edu.ec

endogenous retroviruses typically comprise 5–10% of the whole material (*Waterston et al., 2002*; *Mager & Stoye, 2015*), while about 8% of the human genome is derived from retroviral sequences (*Johnson, 2019*). Recent studies have found endogenous retroviruses in birds, reptiles, amphibians, and fish (*Xu et al., 2018*). Thus, more than 100,000 retroviral elements in humans, 30,420 in birds, and 2,300 in *Xenopus tropicalis* have been identified (*Chalopin et al., 2015*; *Naville & Volff, 2016*).

Retroviruses are made up of four main genes: *gag, pro, pol,* and *env*. Additionally, both exogenous retroviruses and ERVs have long terminal regions (LTRs) that flank their genomes (*Gifford et al., 2018*). The *gag* genes code for specific antigen proteins, including the matrix (MA), the capsid (CA), and the nucleocapsid (NC). The *env* genes encode retroviral envelope proteins, such as the surface protein (SU) and the transmembrane subunit (TM) (*Jern et al., 2005*; *Chen & Cui, 2019*). The *pol* genes are the best-conserved ones through the ERVs and encode a reverse transcriptase (RT) and an integrase (IN) (*Hayward, 2017*). However, these endogenous retroviral elements lose their identity over time, essentially by mutation or recombination, and in some cases become undetectable from their original state (*Diehl et al., 2016*).

ERVs, as well as other transposable elements, play a fundamental role in the vertebrate's evolution (*Biémont, 2010*), and have a strong influence on the host genomes (*Frank & Feschotte, 2017*). It is presumed that an ERV's presence allows new proviral structures to be replicated and inserted, directly affecting the host genome. They could also disrupt the regulation of adjacent genes, as well as promote viral gene expression with long-term genomic effects (*Villesen et al., 2004*; *Mourikis, Aswad & Katzourakis, 2016*). This effect is clearly observed with syncytin, an essential viral protein expressed during placental formation (*Chuong, Tong & Hoekstra, 2010*).

The activity of LTRs in the host genome has often been described as neutral. However, it is likely that the activation of these sequences has a significant impact on the regulatory network of the host genes, because these have retroviral promoters and enhancer elements that can influence the transcription of adjacent genes and the ERVs themselves (*Griffiths, 2001*; *Johnson, 2019*). For example, the disruption of the human ERV-K interferes with cell expression processes, for instance RNA-binding and alternative splicing (*Ibba et al., 2018*). Cancer cell formation has been associated with the ectopic activation of ERVs, presumably due to the stochastic effect of oncogene expression by LTR elements (*Chuong, 2014*; *Pontis et al., 2019*).

In order to differentiate the ERVs from the native genomic signatures, several programs have been developed. Software, such as RetroTector, estimates the probability that a sequence comes from a retrovirus based on a combination of heuristic algorithms (*Sperber et al., 2009*; *Hayward, Grabherr & Jern, 2013*). The main algorithm is based on "chunk threading". First, candidates are selected if LTRs are present. Then, conserved retroviral motifs are detected. Finally, it rebuilds the four major retroviral proteins, namely *gag, pro, pol*, and *env* (*Sperber et al., 2007*). Other programs based on the identification of LTR-RT sequences, such as LTR_FINDER (*Xu & Wang, 2007*), predict the locations and structure of full-length LTR retrotransposons from DNA sequences. LTR-Harvest (*Ellinghaus, Kurtz & Willhoeft, 2008*), for the *de novo* detection of full-length LTR retrotransposons, provides

annotations based on length, distance, and sequence motifs of the LTR retrotransposon. LTR_Retriever (*Ou & Jiang, 2018*) is a multi-threaded program that identifies LTR-RT and generates high-quality LTR libraries from genomic sequences. These programs are essential for obtaining a high-quality genetic annotation; however, they are associated with low specificity rates and high false discovery rates (*You et al., 2015*).

In this work, we traced the evolutionary history of conserved intronic human ERVs and analyzed their potential impact on their host. To do that, we studied the presence and absence of ERVs in the orthologs and paralogs, analyzing 118 vertebrate genomes. Finally, we detected the transcription factors associated with ERV regions to deduce the potential role of these sequences inside the host genomes. With this work, we aimed to shed some light on the role of ERVs in the human genome and understand the causes that lead to the host genome retention of viral information through millions of years of evolution.

## METHODS

### ERV detection in humans

We searched a *de novo* list of human ERVs using LTR-HARVEST, with the same parameters suggested by the authors (*Ellinghaus, Kurtz & Willhoeft, 2008*). Then, the results were filtered with LTR-RETRIEVER (*Ou & Jiang, 2018*) using default parameters. The script call_seqbylist.sh included in LTR-RETRIEVER was used to retrieve the predicted ERVs from a GFF3 file. To validate our results, a BLAST search against the Genome-based Endogenous Viral Element Database (gEVE, http://geve.med.u-tokai.ac.jp/) was performed. Only query sequences whose best hit had a low e-value (1e−5) were considered human ERVs (Table S1). We used RetroTector© (*Sperber et al., 2009*) to perform a second round of filtering to obtain only ERVs with detectable elements of *pol*, *gag*, and *env* genes. Because we focussed on intronic ERVs, we retained candidates located exclusively between exons.

### ERV detection in mammal orthologs and human paralogs

Orthologs for 118 different species were identified using the Orthologous Mammalian Markers (OrthoMam) database (*Ranwez et al., 2007*), based on the list of genes obtained in the previous section as a query. Each ortholog was analyzed with RetroTector to detect the presence of ERV elements. A pairwise alignment was performed for each human ERV candidate and every ortholog detected. We used pairwise BLASTN (*Johnson et al., 2008*) to achieve this task. In addition, the predicted human ERVs were mapped to each ortholog using the Geneious Prime mapping tool (with default parameters) ("*Geneious Prime R10*").

The human paralog genes were identified with Ensembl (*Howe et al., 2021*). The presence of ERVs was evaluated in the same way as the orthologs. Moreover, a multiple alignment was made using the MAFFT (FFT-NS-1 algorithm) (*Katoh & Standley, 2013*) in order to identify paralogs with similar Intron-Exon structures to the target gene.

### Species tree reconstruction

To infer a species tree and reconstruct the evolutionary history of our ERV candidates, we used the *INPP5B* gene, because this is considered a remarkable phylogenetic marker for mammals, in agreement with OrthoMam (*Ranwez et al., 2007*). The species tree was

reconstructed in PhyML (*Guindon et al., 2010*) using the GTR model, allowing the program to estimate the value of the gamma parameter and proportion of the invariable sites. SH-like was used as a supporting measurement of the branches.

We used the presence/absence of ERVs in target genes to estimate the minimum sites of ERV introduction in the species tree.

## Transcription factor binding sites in ERVs

The Gene Transcription Regulation Database (GTRD) was used to determine the transcription factors present in the regions annotated as ERVs. GTRD is based on the BioUML platform, which uses a ChIP-seq data collection of *Homo sapiens, Mus musculus*, and *Rattus norvegicus*, among others (*Yevshin et al., 2019*). A list of annotated transcription factor binding sites was collected for the five genes selected in humans and their orthologs in *Mus musculus* and *Rattus norvegicus*, as well as paralogs in humans with similar structures (introns and exons with the same disposition and length proportion). Once the different lists were obtained, a comparison (presence/absence) was made between orthologs and then between paralogs to determine transcription factors able to bind to the sections specifically annotated as ERV. In addition, an enrichment analysis was carried out in David Bioinformatics (*Huang, Sherman & Lempicki, 2009*) with the transcription factors exclusively found in ERVs to determine the biological processes and metabolic pathways enriched in these regions. Equivalent information was obtained for the paralogs.

Furthermore, an interaction analysis was carried out in BioGrid (*Oughtred et al., 2019*) for the five human genes annotated as intron ERVs. Once the lists were obtained, the interaction networks were generated in the Cytoscape software (*Shannon et al., 2003*). Finally, an enrichment analysis was performed using BiNGO (*Maere, Heymans & Kuiper, 2005*) to determine the gene ontology (GO) categories overrepresented in a set of genes or in a subgraph of an interaction network.

## Analysis of clinical variants

We searched for clinical variants reported in Ensembl (*Howe et al., 2021*) that were identified in the gene sections annotated as ERVs in the human genes. The same analysis was performed for the orthologs and paralogs with available information in Ensembl.

## ERV expression

To regard the expression of ERV in human tissues, we used the information compiled in the UCSC browser (*Kent et al., 2002*) from the GTEx portal on 03/23/2022 (https://gtexportal.org/). The expression patterns observed were limited to the specific genomic coordinates where the ERVs were predicted.

## RESULTS

### Retroviral elements

We performed a thorough analysis of human intronic ERVs with a conserved retroviral structure by applying a *de novo* search. We detected 28 intronic ERVs but only six (inside five human genes) with elements of *pol* and *gag* genes, and in some cases *env* and *pro* genes.

These were: 5′LTR, Primer Binding Site (PBS), Matrix protein (MA), Capsid protein (CA), Nucleocapsid protein (NC), Reverse Transcriptase (RT), Integrase (IN), Viral Protease (PR), Surface envelope protein (SU), Transmembrane protein (TM), Polypurine Tract (PPT), and 3′LTR. Figure 1 shows the ERVs' relative position to the gene and the retroviral elements detected by RetroTector. In all the cases, the ERVs show antisense orientation relative to the transcriptional direction of the gene. The other 22 predicted ERVs have only LTR signals but no recognizable retroviral genes; for that reason, we decided to focus only on the ERVs with the most complete structures.

## Identification of ERVs in orthologs

We analyzed the orthologs of the human genes described in the previous section. 118 orthologs were identified for *INPP5B* and *DET1*, 117 for *PSMA1*, 114 for *USH2A*, and 111 for *MACROD2*. We identified ERV presence in these genes (Table 1) by pairwise alignment in BLASTN and posterior analysis with RetroTector. To consider the genes as ERVs, we took into account the similarity with the human candidates and the identification of ERV elements by RetroTector.

From these analyses, we found evidence of ERV homologs in primates (Tables S2–S7), showing high similarities with human ERVs that have been detected by RetroTector. Outside the primate group, only in some species has the presence of ERVs been detected, and never by the BLAST search and RetroTector at the same time (Figs. S2–S6).

## Paralog identification

We identified 13 paralogs for human *INPP5B*, 19 for *PSMA1*, 28 for *USH2A*, 2 for *MACROD2*, and none for *DET1*. We analyzed the sequences in RetroTector and performed a BLASTN *vs* ERV pairwise alignment for each gene. From the 13 paralogs detected for *INPP5B*, we identified only one sequence in RetroTector for the gene *SYNJ1* (Table S8). However, the query coverage with the *INPP5B* ERV was 4%, so it is likely that it is a different ERV or a fraction of it. We determined 4 genes with signals of retroviral elements in *USH2A* paralogs: *LAMB2, LAMA2, NTN4,* and *TMEFF2* (Table S9). Nonetheless, their alignments show that the retroviral structures are different from those found in human genes (query coverage from 3 to 8%), so we assume that these are different ERVs or remnants of the query retroviral sequence. For *PSMA1* and *MACROD2* paralogs, both RetroTector analysis and the alignments gave negative results, so there is no evidence of retroviral elements in any of their paralogs.

From these results, we considered that there is not enough information to assure the presence of conserved ERVs in the paralogs of the human genes that we analyzed.

## Evolutionary history of ERVs in primates

From the ortholog analysis, we deduced that in order to explain the presence of the ERV observed in *PSMA1,* the viral infections must have occurred at the very least in the common ancestors of the subfamily Homininae. In the case of *DET1*, the retroviral insertion could have taken place at least in the superfamily Hominoidea lineage *(Homo sapiens, Pan troglodytes, Pan paniscus, Pongo abelii,* and *Nomascus leucogenys*), while insertions observed

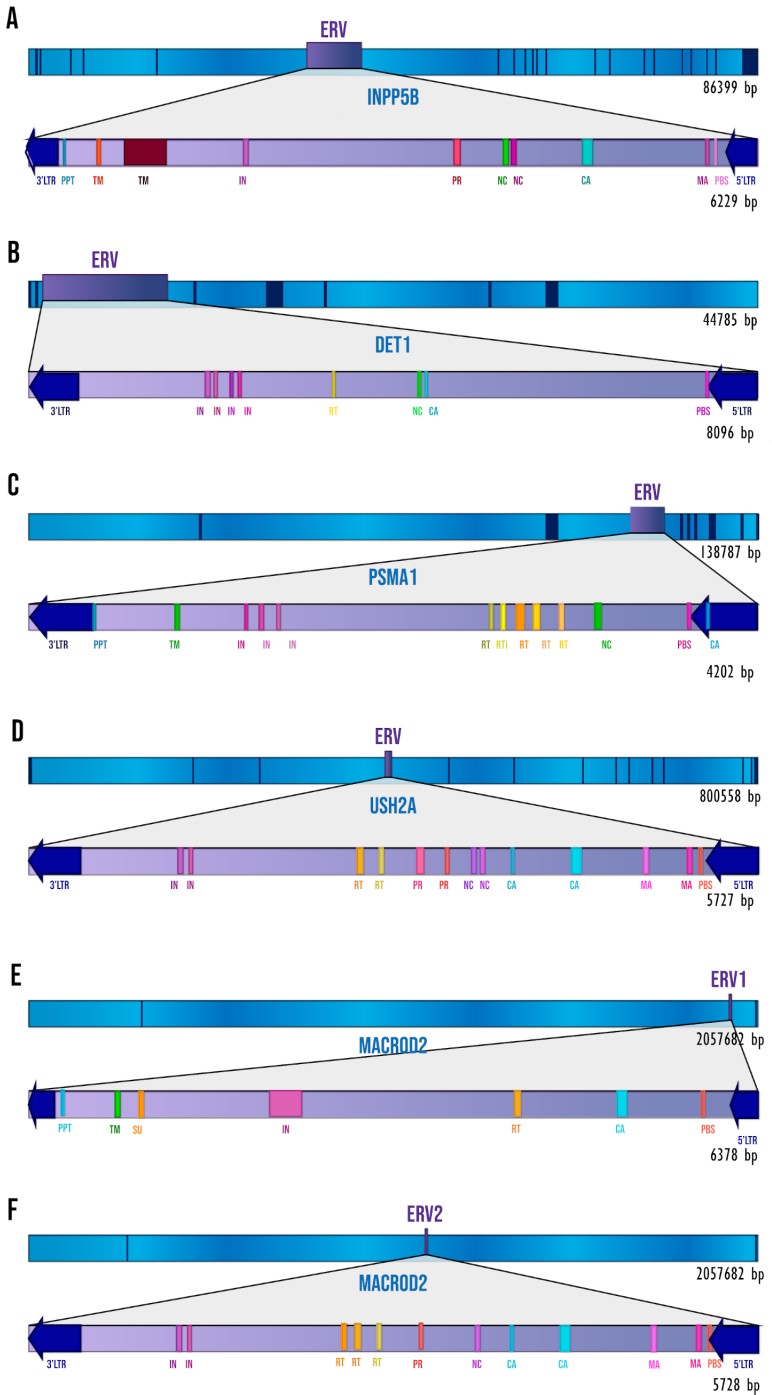

**Figure 1** **The ERVs' structures and locations.** The dark blue segments represent exons, and sky-blue segments represent the introns of each gene. The retroviral elements seen on the diagram are those that have been detected by RetroTector. (A) INPP5B gene. (B) DET1 gene. (C) PSMA1 gene. (D) USH2A gene. (E) MACROD2 gene, ERV1. (F) MACROD2 gene, ERV2.



**Table 1** Candidates' orthologs with ERVs detected in them.

| Human genes with intron ERVs | Number of orthologs identified | ERV detected by RetroTector | ERV detected by pairwise BLASTN[a] |
|---|---|---|---|
| *INPP5B* | 118 | 25 | 15 |
| *DET1* | 118 | 7 | 5 |
| *PSMA1* | 117 | 5 | 4 |
| *USH2A* | 114 | 48 | 17 |
| *ERV1 MACROD2* | 111 | 65 | 13 |
| *ERV2 MACROD2* | | | 3 |

**Notes.**

[a]The presence of an ERV was assumed in sequences with a pairwise alignment similarity higher than 80% and a query coverage higher than 50%.

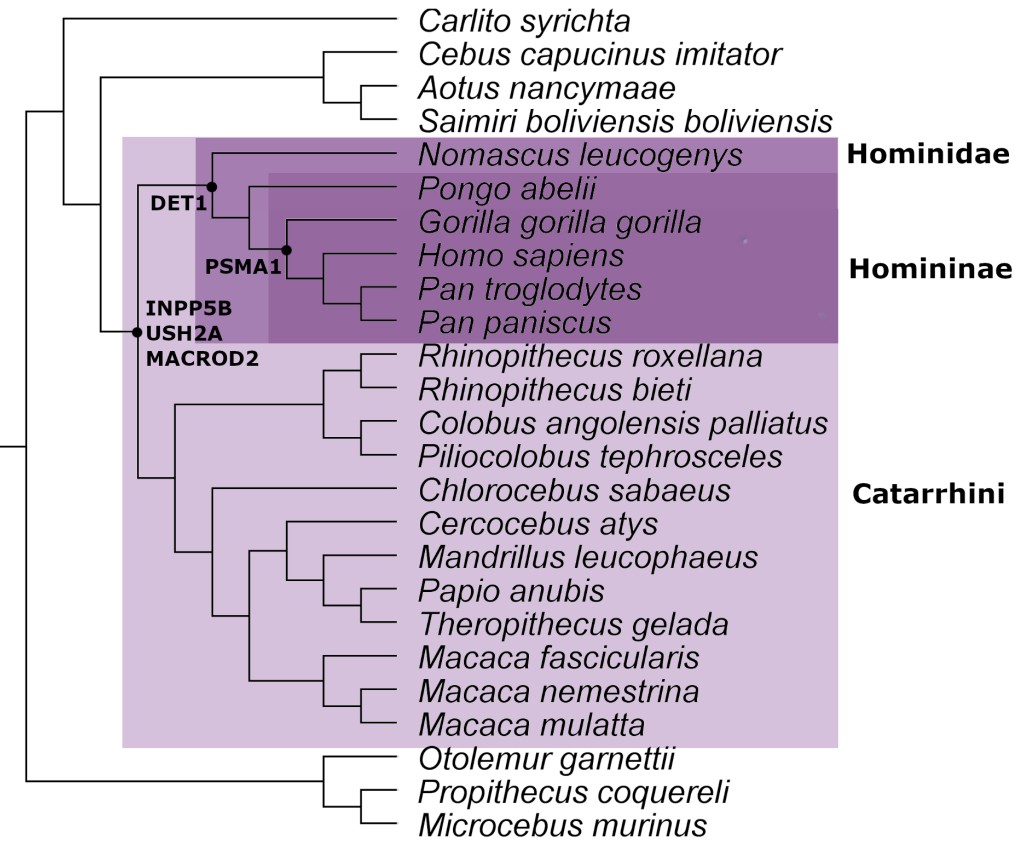

**Figure 2** Mammal species tree with the location of the minimum common ancestor where the retrovirus infection occurred for the genes analyzed in this work.

in *INPP5B*, *USH2A*, and *MACROD2* could, as a minimum, have occurred in the parvorder Catarrhini (Old World Monkeys) (Fig. 2).

This evidence leads us to think that the retroviral infection occurred at least between 35 and 40 million years ago during the evolution and differentiation of Catarrhini primates

(*Aiewsakun & Katzourakis, 2015*). The phylogenetic tree for each gene analyzed is available in Figs. S1–S5.

## Clinical variants associated with ERVs

For the clinical variants annotated in our genes, we described the number and type of variants found in the ERV region of each gene (Table 2). We performed this analysis to explore the potential consequence of our list of intronic ERVs in the host genome. We collected information from: (a) single-nucleotide polymorphism (SNP), defined as a variant that affects a single base pair (although there are multi-allelic SNPs) that must be present in more than 1% of the population; (b) insertion-deletion mutations (indels), which are events where the insertion and/or deletion of less than 1 kb of nucleotides occurs (*Sehn, 2015*); (c) single nucleotide variant (SNV), which like SNPs affect a nucleotide, but this type of mutation is rare and is present in less than 1% of the population (*Harel et al., 2015*); and (d) structural variation that is considered a large rearrangement, produced by deletions, duplications, insertions, inversions, or translocations (*Freeman et al., 2006*). All these mutations can occur in both somatic and germline cells.

Furthermore, we collected information about somatic type variants, related to different types of tumors. In the case of the ERV region of *INPP5B*, we found variants associated with tumors in the esophagus (one), breast (three), pancreas (one), liver (seven), hematopoietic and lymphoid tissue (four), ovary (six), prostate (two), and kidney (one). For *DET1*, we found somatic variants related to tumors in hematopoietic and lymphoid tissue (two), the upper aerodigestive tract (one), the esophagus (one), prostate (one), breast (two), liver (one), kidney (two), and the central nervous system (one). In *PSMA1* ERV, we found somatic variants associated with breast (five), stomach (one), prostate (two), liver (one), hematopoietic and lymphoid tissue (six), large intestine (one), and esophageal (one) tumors. In the *USH2A* ERV region, three variants have been reported—2 SNPs and one INDEL; we also discovered two variants linked to regulatory activity—one variant in the TF binding site and one in the CTCF binding site.

Only one somatic SNV associated with kidney tumors has been reported in the ERV1 *MACROD2* region (Table S10). Regarding the ERV2 region, there is no information on clinical variants, but there are four variants in regulatory regions: one in a promoter, one in an enhancer, and two copy number variations (CNV).

## Transcription factors associated with ERVs and intronic ERV expression

Transcription factor binding sites (TFBSs) have been observed in association with HERVs and LTRs, regulating the transcriptional activity of these exogenous elements (*Monde et al., 2022*; *Liu et al., 2022*). In order to determine the transcription factors (TFs) associated exclusively with ERVs, we excluded all the TFBSs shared by orthologs and paralogs annotated in structural equivalent sections. We obtained 28 TFBSs for *INPP5B*, 188 for *DET1*, 58 for *PSMA1*, 19 for *USH2A*, 96 for ERV1 *MACROD2,* and 238 for ERV2 *MACROD2* (Table S11).

We made an enrichment analysis with David Bioinformatics Resources to find the pathways in which these factors are involved, and how many of these are related to cancer.
**Table 2  Number of clinical variants annotated in the ERV regions of five human genes.**

| Gene | Variants | 1,474 | Gene | Variants | 3 |
|------|----------|-------|------|----------|---|
| | SNP | 1347 | | SNP | 2 |
| | INDEL | 81 | | INDEL | 1 |
| | Deletion | 19 | | Deletion | – |
| INPP5B | Insertion | 2 | USH2A | Insertion | – |
| | Somatic SNV | 24 | | Somatic SNV | 1 |
| | Somatic deletion | 1 | | Somatic deletion | – |
| | Somatic insertion | – | | Somatic insertion | – |
| | Regulatory activity variants | – | | Regulatory activity variants | 2 |
| Gene | Variants | 1041 | Gene | Variants | 1 |
| | SNP | 938 | | SNP | – |
| | INDEL | 67 | | INDEL | – |
| | Deletion | 21 | | Deletion | – |
| DET1 | Insertion | 4 | ERV1 MACROD2 | Insertion | – |
| | Somatic SNV | 9 | | Somatic SNV | 1 |
| | Somatic deletion | 2 | | Somatic deletion | – |
| | Somatic insertion | – | | Somatic insertion | – |
| | Regulatory activity variants | – | | Regulatory activity variants | – |
| Gene | Variants | 1121 | Gene | Variants | – |
| | SNP | 1038 | | SNP | – |
| | INDEL | 49 | | INDEL | – |
| | Deletion | 14 | | Deletion | – |
| PSMA1 | Insertion | 3 | ERV2 MACROD2 | Insertion | – |
| | Somatic SNV | 15 | | Somatic SNV | – |
| | Somatic deletion | 1 | | Somatic deletion | – |
| | Somatic insertion | 1 | | Somatic insertion | – |
| | Regulatory activity variants | – | | Regulatory activity variants | 4 |

Table 3 summarizes these results. Additionally, we performed the same analysis with the paralogs (using structural equivalent regions to ERVs). We observed that only three of the 13 sections analyzed in paralogs were enriched in TFs associated with cancer, in contrast with five out of six that are enriched for the ERVs (Table S12).

Despite the position of ERVs inside introns, some of the candidates have a low but visible expression in certain tissues and cells. In the case of INPP5B, a high expression peak is observed in brain tissues in a short section around the transmembrane gene of the ERV (163 bp). Remarkably, no other section of the ERV has an expression signal in this case. Intrigued by this observation, we used this section to find copies throughout human genomes, and we found more than 10 copies located inside the introns of several genes with a high expression exclusively in brain tissues (File S2). Additionally, several variants associated with tumors within these sections of high expression were observed (Table S13). We also found copies of this section in intergenic regions but without expression in any tissue.
**Table 3 ERV transcription factor enrichment analysis.**

| Gene name | No of unique transcription factors associated with ERVs | Statistically significant enriched KEGG pathways | Cancer Pathways | Transcription factors involved in Cancer Pathways | Benjamini[*] |
|---|---|---|---|---|---|
| INPP5 | 28 | 0 | – | – | – |
| DET1 | 188 | 5 | Transcriptional misregulation in cancer | 28 | 8.40E−13 |
| | | | Pathways in cancer | 18 | 3.10E−04 |
| | | | Viral carcinogenesis | 12 | 1.60E−03 |
| PSMA1 | 58 | 1 | Transcriptional misregulation in cancer | 9 | 1.00E−05 |
| | | | Acute myeloid leukemia | 4 | 2.20E−03 |
| USH2A | 19 | 3 | Transcriptional misregulation in cancer | 5 | 2.20E−03 |
| | | | Pathways in cancer | 5 | 3.80E−02 |
| ERV1 MACROD2 | 96 | 17 | Transcriptional misregulation in cancer | 20 | 4.30E−17 |
| | | | Pathways in cancer | 15 | 1.20E−05 |
| | | | Acute myeloid leukemia | 6 | 7.40E−04 |
| | | | Renal cell carcinoma | 5 | 1.10E−02 |
| | | | Chronic myeloid leukemia | 5 | 1.30E−02 |
| | | | Viral carcinogenesis | 7 | 1.90E−02 |
| | | | MicroRNAs in cancer | 8 | 2.10E−02 |
| ERV2 MACROD2 | 238 | 6 | Transcriptional misregulation in cancer | 5 | 2.70E−02 |

**Notes.**
*Adjusted *p*-values by using the linear step-up method by *Benjamini & Hochberg (1995)*.

The ERV located inside *DET1* has a dispersed pattern of expression in almost all the tissues available in GTEx, especially in the tibial nerves and testes. A similar pattern was observed for the ERV located inside *PSMA1*. Both ERVs located inside *MACROD2* have low levels of expression; however, the ERV2 is particularly expressed in brain tissues (File S3).

## DISCUSSION

In this work, we found retroviral elements in different vertebrate genomes and suggest the implications of the retention of this viral information over millions of years of evolution. From our analysis, a clear pattern of ERV gains and losses was identified in primates. Additionally, we were able to determine the presence of regulatory sequences inside the ERV regions.

Endogenous retroviruses can preserve the *cis-* and *trans-* acting mechanisms of the exogenous virus, which could at any time make it potentially dangerous for the host, even though millions of years have passed since its integration and despite the provirus being severely degenerated (*Lander et al., 2001*; *Blomberg, Ushameckis & Jern, 2013*). In all the ERVs that we found, *gag* and *pol* structures were observed, and the *pro* and *env* domains can be recognized in the ERVs of *INPP5B*, *USH2A*, and *MACROD2* (Fig. 1). Even though ERVs are conserved in these structures, this does not imply that they still have an infectious capacity as a retrovirus (*Jern, Sperber & Blomberg, 2004*; *Marie, Sandra & Thierry, 2005*).
To explore this possibility, we observed the expression of our intronic ERVs (File S3). We found a lack or low levels of expression in most of the ERVs, making it unlikely that the retroviral genetic information still has an infective capacity. A remarkable exception was observed in the 163 bp section inside the *INPP5B* ERV. After several BLAST searches, we annotated this section as an antisense long non-coding RNA (NONHSAG056331.1), in agreement with the NONCODE database (*Zhao et al., 2021*) (http://www.noncode.org/). The 163 bp section is included in the 917 bp length NONHSAG056331.1; nevertheless, the expression of the entire lncRNA is higher in the lymph nodes (a tissue not available in GTEx data) than in the brain. We found at least 10 human genes with a very similar sequence to the 163 bp section (with a similarity of more than 92%) within introns and highly expressed in brain tissues (according to the GTEx data) (see File S2). The 163 bp section was not annotated as part of an ERV gene in RetroTector, but searches in RepeatMasker (http://www.repeatmasker.org) and RepBase (*Bao, Kojima & Kohany, 2015*) annotated this sequence as an LTR and ERV class 1. Although we found several ERV structures inside the *INPP5B* intron, the evidence indicates that it is unlikely that the entire ERV is expressed. However, the 163 bp section deserves to be studied further to determine the cause and consequences of its expression.

ERVs, after their insertion into the host genome, can undergo recombination events or accumulate multiple mutations (*Löber et al., 2018*; *Halo et al., 2019*), leading to their inactivation, although some ERVs retain their ability to replicate (*Kozak, 2015*). Published methods for the retrieval of retroviral sequences from genomic databases focus on long terminal repeat pair detection, specific conserved sequences, or general repeat detection (*Steinbiss et al., 2009*; *Shi & Liang, 2019*). These limitations in detection methods may result in false negatives or false positives in ERV research, making it even more difficult to determine the presence or absence of ERVs in certain genomes. In the same way and due to these limitations, it is likely that the LTRs changed their sequence and avoided being recognized by RetroTector and some other software based on LTR recognition. In this work, we found several genes in vertebrates without ERV signals (in agreement with RetroTector); however, BLAST alignments showed a high similarity with the human ERVs. Using a strict criterion, we do not consider these sections to be ERVs, but it is plausible that these genes already contain ERVs with one or more of the retroviral signatures deleted. We compiled this information in the trees presented in Figs. S1–S6.

Various methods are used to determine the time interval in which a retrovirus infects a germline in a host species to lead to the appearance of an ERV. One of these is by determining the presence or absence of an ERV in the genomes of phylogenetically related species (*Johnson, 2015*). In general, if an infection occurred in a recent common ancestor, all species or most of the descendants must retain the ERV. If the retrovirus infection occurred in an ancient ancestor, more losses are expected. This robust method only provides an estimation of the interval in which the infection could occur (*De Parseval & Heidmann, 2005*; *Hron et al., 2016*). Using this approximation, we estimated the minimum ancestor for each ERV appearance analyzed in this work (Fig. 2). It is important to remark that we found evidence of ERVs in several species of primates. This level of conservation points out

the relevant role of the ERVs inside their hosts, since the retention of the ERVs by genetic drift in up to 18 species is highly unlikely.

Our results show that the ERV integrations occurred along the Catarrhini lineage, made up of hominoids and cercopithecoids (*Groves, 2016*). These two superfamilies diverged approximately 32 million years ago (MYA) according to fossil and genetic evidence (*Pozzi et al., 2014*). In a similar experiment, *Vargiu et al. (2016)* estimated the origin of 3,173 human ERVs (HERVs) from six to 100 MYA. This implies that the integration took place after the *Eutheria* divergence but before the differentiation between chimpanzees and humans. In the same line, *Grandi et al. (2016)* performed a phylogenetic and structural analysis of HERV-W (a group of human endogenous retroviruses widely studied due to their participation in various diseases), based on the divergence rate of the nucleotides. These results date the acquisition of this element during the Catarrhini lineage evolution (40 and 20 MYA approximately), just after the separation of the parvorder Platyrrhine. These results coincide with our estimations, suggesting that the parvorder Catarrhini could be a hotspot of ERV acquisition, and perhaps these external elements contributed toward shaping the evolutionary pathway of this lineage.

We found a cluster of TFBSs annotated in ERV regions and enriched in TFs associated with transcriptional misregulation in cancer. This observation raises the question of whether ERV regions tend to accumulate TFBSs or if they are independent of the exogenous material. A similar observation has been reported for p53 TFBSs, where 1,509 LTR of human ERVs have a p53 DNA binding site (*Wang et al., 2007*). In addition, the clinical variants annotated in the ERV regions coincide with variants associated with tumors, reinforcing the idea that the analyzed ERV sections have important roles in cell cycle regulation and their misregulation leads to cancer. In agreement with these observations, several authors have described an association between ERVs and cancer; moreover, this relationship is explained by ERV activation (*Ibba et al., 2018*; *Topham et al., 2020*). Following our data, another possibility for disrupting the cell cycle and producing tumors could be through TFBS mutations that prevent TFs from binding to DNA. The regulatory regions observed in the ERVs analyzed in this work are absent in paralog sequences without ERVs. Do the ERVs facilitate the host's gene regulation?

In addition to the association of ERVs with cancer development and other diseases, there is evidence that these retroviral sequences could play a role in immune responses, placental development, and so on (*Bannert et al., 2018*). The best-known examples of functional proteins produced during placentation are Syncytin-1 and Syncytin-2, which are critical in underlying cell fusion for the formation and maintenance of the placenta (*Chen et al., 2008*). Furthermore, the ERV sequence functions as an immune functional unit. So, this can serve as an antiviral sequence that allows the inhibition and destruction of foreign DNA when a viral infection occurs with a similar sequence (*Hammen, 2018*). *Chiappinelli et al. (2015)* demonstrated that when the ERV bidirectional transcription occurs, the type I interferon response is triggered and apoptosis of the infected cell occurs through the activation of a double-stranded RNA detection pathway.

Another interesting question related to the transcription factor analysis is whether the transcription factor binding sites can activate or deactivate ERVs (*Grow et al., 2015*;

*Monde et al., 2022*). A frequent target of ERV silencing is the so-called primer binding site (PBS), although there are other described mechanisms associated with this process. It has been shown that the transcription factor TRIM 28 is involved in proviral silencing mechanisms (*Geis & Goff, 2020*). This process is believed to depend on the glycosylation of the protein (*Rowe et al., 2010*; *Boulard et al., 2020*). Scientists have even described how the deglycosylation of these proteins can reactivate the transcription of methylated retrotransposons promoters (*Rowe et al., 2010*). Within our analysis, we found that the TRIM 28 factor was present in all the studied ERVs, and moreover, other TRIM family members were found in two ERVs. For the moment, we have not recovered any more information to suggest a relevant role of this family of transcription factors in ERV activation. Other important TFBSs observed in our ERVs include the ZNF (Zinc Finger) protein family, which binds to ERVs by a sequence-dependent mechanism, thus potentially participating in the regulation of these viral sequences (*Rajagopalan & Jha, 2018*). In our analysis, around 19 members of this family were present in at least 1 or 2 of the ERVs that were the subject of this investigation. The bromodomain family proteins (BRD) are believed to recruit other complexes to activate or repress gene expression (*Frank et al., 2003*). In our results, we found binding sites to Bromodomain-containing protein factors 2, 3, 4, and 9. Another relevant TFBS found in this study was that which is associated with TIP60. This molecule is capable of silencing ERVs in the presence of BRD4 (*Rajagopalan et al., 2018*). The complete list of TFBSs is available in Table S9.

Despite the clinical variants and transcription factors associated with the ERV regions, at this point we are not able to formulate a strong hypothesis for the role of these exogenous viral materials inside their hosts.

## CONCLUSIONS

We developed a battery of experimental procedures to elucidate the role of ERVs described in this work. The main questions that arise from our results are as follows. How has exogenous genetic material been conserved in primates' genomes (particularly in introns) over millions of years? Do these retroviral elements have an increased capacity to regulate genes or do they have some other unveiled role in the evolution of the genomes?

We were able to trace the evolutionary history of six ERVs throughout primate evolution. The ancientness deduced for the retroviral sequences led us to think that these ERVs survived by natural selection and were co-opted to perform certain roles for their host, as per the list of 93 vertebrate ERVs reported by *Wang & Han (2020)*. This work proposes new questions surrounding the function and evolution of ERVs, suggesting a relevant role of these exogenous elements within their hosts.

## ACKNOWLEDGEMENTS

We would like to thank Helen Pugh for proofreading the manuscript.

### Funding

This work was supported by Universidad de Las Américas-Ecuador, project BIO.VAJ.19.06. The funders had no role in study design, data collection and analysis, decision to publish, or preparation of the manuscript.

### Grant Disclosures

The following grant information was disclosed by the authors:
Universidad de Las Américas-Ecuador: BIO.VAJ.19.06.

### Competing Interests

The authors declare there are no competing interests.

### Author Contributions

- Melissa Calero-Layana performed the experiments, analyzed the data, prepared figures and/or tables, authored or reviewed drafts of the article, wrote the first manuscript, and approved the final draft.
- Carmen López-Cruz performed the experiments, prepared figures and/or tables, authored or reviewed drafts of the article, wrote the first manuscript, and approved the final draft.
- Agustín Ocaña performed the experiments, authored or reviewed drafts of the article, collect primary data, and approved the final draft.
- Eduardo Tejera conceived and designed the experiments, analyzed the data, authored or reviewed drafts of the article, designed the first experiments, and approved the final draft.
- Vinicio Armijos-Jaramillo conceived and designed the experiments, performed the experiments, analyzed the data, prepared figures and/or tables, authored or reviewed drafts of the article, wrote the first manuscript, and approved the final draft.

### Data Availability

The raw data is available in the Supplementary Files.

### Supplemental Information

Supplemental information for this article can be found online at http://dx.doi.org/10.7717/peerj.14431#supplemental-information.

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
