# Peer review of "Evolutionary analysis of endogenous intronic retroviruses in primates reveals an enrichment in transcription binding sites associated with key regulatory processes"

_PeerJ, doi:10.7717/peerj.14431_

## Round 0.1 · original submission · Major Revisions

· Academic Editor

Major Revisions

Dear Dr. Calero-Layana and colleagues:

Thanks for submitting your manuscript to PeerJ. I have now received two independent reviews of your work, and as you will see, the reviewers raised some minor concerns about the research (mostly the manuscript format and content). Despite this, these reviewers are optimistic about your work and the potential impact it will have on research studying primate retroviruses. Thus, I encourage you to revise your manuscript, accordingly, taking into account all of the concerns raised by both reviewers.

While the concerns of the reviewers are relatively minor, this is a major revision to ensure that the original reviewers have a chance to evaluate your responses to their concerns. There are not too many suggestions; thus, it should not take much effort to address these concerns to greatly improve your manuscript.

I look forward to seeing your revision, and thanks again for submitting your work to PeerJ.

Good luck with your revision,

-joe

Reviewer 1 ·

Basic reporting

This manuscript by Calero-Layana et al. describes the evolution of intronic ERVs in primates and the enrichment of transcription factor binding sites in/near those ERVs. The authors (1) use clear and unambiguous professional language throughout with only minor necessary changes, (2) provide sufficient background with references, (3) maintain a conventional paper structure with raw data shared, and (4) report a self-contained story with hypotheses.

Experimental design

The authors studied the ERVs in orthologous/paralogous genes and their conservation in vertebrates using publicly available database sequences and data. The research question is defined as attempting to shed light on the potential role of ERVs in the human genome and causes of their retention over evolutionary time. The work is rigorous/technical/ethical and with methods described to PeerJ standards.

The work is bioinformatic in nature and is appropriately modest in its claims, it is a nice example of using publicly available data to inform evolutionary relationships and provide examples of genes or genomic segments that are maintained in lineages over millions of years. The questions posed are interesting regarding what the function (if any) of these intronic ERVs may be.

Validity of the findings

The findings appear valid, and conclusions are well stated with exciting new questions stemming from their work.

Additional comments

1. Figures S2-S6 I would suggest different colors or ways of differentiating species or dual coding. It looks like some species names are green (e.g. Monodelphis domesticus) in Figure S2. If I’m mistaken (I'm red/green colorblind), that’s fine but this could also just be a simple change needed (i.e. the legend should say those “species detected by RetroTector only in green”).
a. Thank you for discussing these non-primate non-ERVs in your discussion with explanation. Would you speculate that perhaps these syntenic regions are prone to integration but germline infection and integration is rare?

2. Between the paragraphs ending at line 196 and starting at 198 (Clinical variants associated with ERVs), please provide a rationale or reminder of the rationale for why you expected Clinical variants OR not and how this relates to the work above (finding ERVs in these genes). Are these genes important in cancer or development or disease? Is there something shared about the function of the five genes?

3. Similarly, paragraph starting at line 222 the authors state, “In order to determine the transcription factors…”. Please state why you expected TFBS in these regions, for example, “Because the clinical variants found within these ERVs, associated with multiple tissues, and some of the variants occurring at transcriptional regulators, we wondered if these ERVs were associated with TFBS.”

4. Adjust the language at lines 160-161 “where was finded”, I suggest “ERVs show antisense orientation relative to the transcriptional direction of the gene.”

Reviewer 2 ·

Basic reporting

- In the introduction when “genome” is referenced it should be specified if it is the viral genome or host genome. In many instances it is not clear which is being referenced.

- In general, the manuscript has grammatical issues and spelling errors throughout, these should be corrected.

Experimental design

- The introduction lists several software programs that can be used for ERV detection with a few listed caveats. However, it is not clear why RetroTector was utilized for this study. Is RetroTector superior to the other programs? Does it not have these same caveats?

Validity of the findings

- Do any of the clinical variants associated with ERVs occur at a frequency that is statistically significant relative to genes not associated with ERVs? For example, the authors state that INPP5B has ~1350 SNPs in the gene, with ~25 of those occurring in tumors for a frequency of ~2%. Is this frequency much higher relative to other genes that have a similar number of SNPs that are not associated with ERVs? This in an important distinction for interpreting if ERV integration has a potential impact on clinical outcomes.

- In addition to the point above, are the somatic type variants referenced in the text different variants, or are they the same variant just in a different cancer context?

-For the TFBS analysis in Table 3, the authors state that the TFs associated with ERV-associated genes are significantly enriched for pathways involved in cancer. However, this analysis is only focused on ERV-associated genes. If this analysis is performed using TFBSs for structurally similar genes without ERV integration, is the opposite trend observed (i.e., no significant enrichment, because the ERV is driving this phenotype)?

-For instances where a specific ERV was determined to have multiple integration sites in different genes, were there any disease-associated somatic type variants in these genes? For example, on line 235 it states that “we found more than 10 copies located inside introns of several genes with a high expression…” Do these genes have tumor-associated SNPs?

-The results section states that only 6 ERVs were discovered inside of 5 human genes, however roughly 30 genes are listed in Supp Table 1 as having intronic ERVs. What is the rationale for this discrepancy? Do these not contain all ERV elements? If this is the case, this rationale should be explicitly stated in this paragraph of the results section.

Additional comments

- Line 54, Diehl reference is missing a publication date

---

## Round 0.2 · accepted · Accept

· Academic Editor

Accept

Dear Dr. Calero-Layana and colleagues:

Thanks for revising your manuscript based on the concerns raised by the reviewers. I now believe that your manuscript is suitable for publication. Congratulations! I look forward to seeing this work in print, and I anticipate it being an important resource for groups studying primate retroviruses. Thanks again for choosing PeerJ to publish such important work.

Best,

-joe

Reviewer 1 ·

Basic reporting

No comment. Article as revised meets PeerJ standards.

Experimental design

No comment. Article as revised meets PeerJ standards.

Validity of the findings

No comment. Article as revised meets PeerJ standards.

Additional comments

No comment.

Reviewer 2 ·

Basic reporting

The authors have addressed all of my comments and concerns from my initial review. I have no further comments and/or concerns about the revised study.

Experimental design

The authors have addressed all of my comments and concerns from my initial review. I have no further comments and/or concerns about the revised study.

Validity of the findings

The authors have addressed all of my comments and concerns from my initial review. I have no further comments and/or concerns about the revised study.